# CartoonSing: Unifying Human and Nonhuman Timbres in Singing Generation

## Abstract

Singing voice synthesis (SVS) and singing voice conversion (SVC) have achieved remarkable progress in generating natural-sounding human singing. However, existing systems are restricted to human timbres and have limited ability to synthesize voices outside the human range, which are increasingly demanded in creative applications such as video games, movies, and virtual characters. We introduce Non-Human Singing Generation (NHSG), covering non-human singing voice synthesis (NHSVS) and non-human singing voice conversion (NHSVC), as a novel machine learning task for generating musically coherent singing with non-human timbral characteristics. NHSG is particularly challenging due to the scarcity of non-human singing data, the lack of symbolic alignment, and the wide timbral gap between human and non-human voices. To address these challenges, we propose CartoonSing, a unified framework that integrates singing voice synthesis and conversion while bridging human and non-human singing generation. CartoonSing employs a two-stage pipeline: a score representation encoder trained with annotated human singing and a timbre-aware vocoder that reconstructs waveforms for both human and non-human audio. Experiments demonstrate that CartoonSing successfully generates non-human singing voices, generalizes to novel timbres, and extends conventional SVS and SVC toward creative, non-human singing generation. Audio samples are available at `https://cartoonsing.github.io/`.

## 1 Introduction

Singing voice synthesis (SVS) and singing voice conversion (SVC) have achieved remarkable progress in generating high-fidelity, natural singing voices from symbolic music or reference human singing. Modern approaches demonstrate strong performance in pitch accuracy, rhythmic alignment, timbre similarity, and expressive control (Lu et al., 2020; Liu et al., 2022; Zhang et al., 2023a; Wu et al., 2024; Dai et al., 2024; Cui et al., 2024; Zhang et al., 2024b; Sha et al., 2024; Chen et al., 2024; Zhang et al., 2025b). However, these methods are mainly focused on reproducing human-like timbres, optimizing the minimal perceptual difference from ground-truth audio, both in training objectives and evaluation design (Gupta et al., 2017; Shi et al., 2021; Tang et al., 2024; Narang et al., 2024; Bai et al., 2026).

By contrast, creative domains such as music production, video games, and movies have widely adopted voices that intentionally deviate from natural human realism. Commercial SVS platforms such as VOCALOID have gained widespread use in music production for their electronically stylized timbres that do not resemble natural human voices (Kageyama, 2023; Radulovic, 2025). Video games such as Splatoon use heavily processed, underwater-like vocalizations to create a distinctive cartoonish auditory experience (Sekai Sandai-gawa Editorial Department, 2015; 2019). Similar strategies appear across popular media, from the accelerated, squeaky voices in *The Chipmunk Song* (Cox, 2018), to the robotic voice acting of GLaDOS in *Portal* (Watercutter, 2013), to the animal-inspired design of Grogu's voice in *The Mandalorian* (Johnson, 2019). These cases illustrate a consistent design strategy of adopting voices beyond natural human production, typically realized through manual digital signal processing (DSP)-based sound design or professional voice acting, both of which constrain scalable creative exploration of timbral spaces outside the natural human vocal range.

This mismatch between research objectives and practical creative demands motivates the definition of a new problem setting: Non-Human Singing Generation. Non-Human Singing Generation (NHSG), including Non-Human Singing Voice Synthesis (NHSVS) and Non-Human Singing Voice Conversion (NHSVC), is introduced as a machine learning task of generating musically coherent singing voices whose timbral characteristics are intentionally distinct from human voices while preserving intelligibility, pitch accuracy, rhythmic consistency, and acoustic qualities.

Exploring non-human singing generation brings challenges that do not appear in conventional SVS and SVC. (1) *The main difficulty is data.* While some commercial products such as singing synthesizers or video game voices provide synthetic or non-human singing timbres, these voices are stylistically narrow and not available for research purposes, making it impossible to build a diverse and open dataset for supervised learning. (2) *Alignment and annotation are also nontrivial.* Unlike human singing recordings, which can be naturally aligned with lyrics and musical scores as required for singing voice synthesis, non-human sounds lack such inherent structure. (3) Finally, *the timbral gap between human and non-human audio is significantly wide.* Systems trained only on human data cannot simply transfer in a zero-shot manner. Effective ML approaches must therefore be designed to bridge this gap while preserving the musical and linguistic consistency of human singing.

To address these challenges, we design CartoonSing, a unified two-stage generation framework for NHSVS and NHSVC. The framework factorizes the information required for synthesis into three components: content tokens obtained by applying k-means quantization to self-supervised learning features, embeddings that specify timbre, and fundamental-frequency (F0) contours that provide pitch information. In **Stage 1**, a score representation encoder is trained on human singing data to predict content tokens and F0 contours from symbolic musical inputs. In **Stage 2**, a unified timbre-aware vocoder reconstructs audio waveforms from the predicted tokens, F0 contours, and timbre embeddings and is trained jointly on human singing and non-human audio. This design addresses the absence of explicit annotation in non-human sounds, unifies NHSVS and NHSVC, and supports generalization to non-human timbres.

To assess the quality of the generated audio, we conduct both objective and subjective evaluations. The results show that our approach achieves consistently strong performance and outperforms competitive baselines across evaluations.

In this work, we make the following contributions:

- We introduce and mathematically formulate **Non-Human Singing Voice Synthesis (NHSVS)** and **Non-Human Singing Voice Conversion (NHSVC)** as a machine learning problem, extending traditional SVS and SVC to non-human timbres. This formalization frames NHSVS and NHSVC as a study of timbre generalization and cross-domain singing generation, which allows systematic exploration of model behavior outside the distribution of natural human vocals.

- We propose **CartoonSing**, the first stable framework and training strategy that unifies NHSVS and NHSVC, supporting zero-shot synthesis and conversion of singing voices beyond natural human timbres.

- We conduct a comprehensive evaluation for NHSVS and NHSVC and provide reproducible baselines from open-source audio datasets to facilitate systematic assessment and future research on non-human singing generation.

## 2 RELATED WORKS

Singing voice synthesis (SVS) and singing voice conversion (SVC) have recently advanced toward more expressive and robust human singing generation, with a growing focus on zero-shot and out-of-domain generalization. Existing evaluations primarily examine generalization to unseen singers, unseen styles, or cross-lingual synthesis (Zhang et al., 2024a; Dai et al., 2024; Chen et al., 2024; Zhao et al., 2025; Zhang et al., 2024b; 2025b). Building on this paradigm, recent studies extend SVS to speech-to-singing generation, where models adapt stylistic or timbral information from human speech as conditional inputs (Zhang et al., 2024b; 2025b; Dai et al., 2025). In a related but distinct direction, Vevo2 (Zhang et al., 2025a) investigates humming-to-singing and instrument-to-singing, with non-vocal inputs purely as melodic or prosodic guidance rather than timbral conditioning.

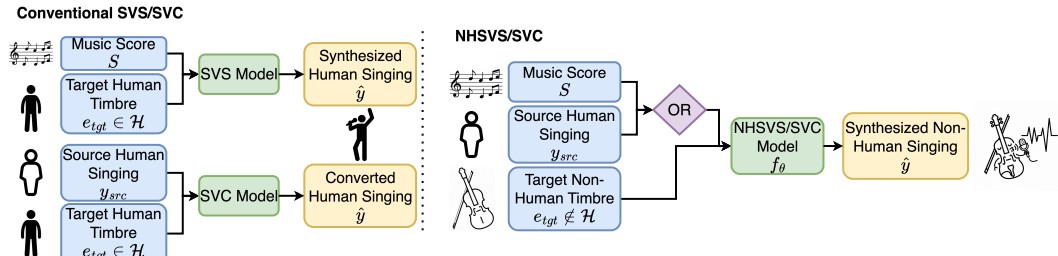

Figure 1: Comparison of task formulations for conventional singing voice synthesis (SVS) and conversion (SVC) versus non-human singing voice synthesis (NHSVS) and conversion (NHSVC).

Meanwhile, some prior work has addressed non-human vocalizations in the speech domain, specifically in the context of voice conversion (VC), relying on cross-domain training with non-human audio, without operating in a zero-shot setting (Suzuki et al., 2022; Kang et al., 2025). Speak Like a Dog (Suzuki et al., 2022) formulates the non-human voice conversion task as class-conditioned VC, using species-level labels to jointly model human speech and dog vocalizations. However, such label-based formulations do not generalize naturally to diverse domains. To address this limitation, Kang et al. (2025) propose a style encoder that extracts reference embeddings in place of explicit labels, combined with single-layer self-supervised learning features to support training across a wider set of non-human sources, including expressive human exclamations, sound-designed characters, and animal sounds. These studies highlight the feasibility of extending voice generation beyond human natural timbres but also reveal unique challenges in linguistic intelligibility, timbral transfer, and zero-shot generalization.

Exploration of singing generation with non-human timbres, however, remains sparse. The only closely related work is SaMoye (Wang et al., 2024), which is trained on large-scale human singing corpora and evaluates zero-shot singing voice conversion (SVC) on a limited set of non-human timbres, specifically five cat and dog timbres. While demonstrating the potential of non-human SVC, its generalizability to broader non-human domains remains unclear. Moreover, SVS presents additional difficulties compared to SVC, as it requires conditioning on musical scores while relying on fewer acoustic cues, making non-human singing synthesis an underexplored and particularly challenging research direction.

## 3 NON-HUMAN SINGING GENERATION

### 3.1 TASK FORMULATION

We formulate Non-Human Singing Voice Synthesis (NHSVS) and Non-Human Singing Voice Conversion (NHSVC) as a conditional generative modeling problem.

For NHSVS, given a symbolic musical score $S$ and a non-human timbre embedding vector $e_{\text{tgt}}$ as reference, we define a conditional generative model $f_\theta$ that produces a waveform $\hat{y}$ preserving the musical content of $S$ while reproducing the timbral characteristics $e_{\text{tgt}}$.

Formally,

$$f_\theta : (S, e_{\text{tgt}}) \mapsto \hat{y}, \tag{1}$$

subject to

$$\mathcal{M}(\hat{y}) \approx S, \mathcal{T}(\hat{y}) \approx e_{\text{tgt}}, \mathcal{T}(\hat{y}) \notin \mathcal{H}. \tag{2}$$

where $\mathcal{M}(\cdot)$ denotes the symbolic musical content of a waveform (e.g., pitch, lyrics, and duration) in the same representation as $S$, $\mathcal{T}(\cdot)$ denotes a timbre embedding function, and $\mathcal{H}$ represents the embedding manifold of natural human singing timbres.

For NHSVC, the score input $S$ is replaced by a source audio waveform $y_{\text{src}}$, from which symbolic musical content is extracted via $\mathcal{M}(y_{\text{src}})$. The model then generates $\hat{y}$ that preserves the musical content of $y_{\text{src}}$ while transferring the target non-human timbre:

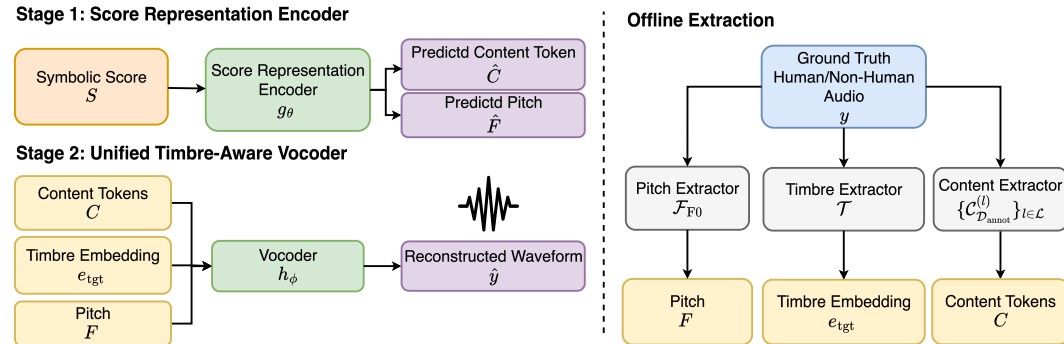

Figure 2: An overview of the proposed two-stage synthesis pipeline. Stage 1 trains a score representation encoder $g_\theta$ on annotated human singing data. Stage 2 trains a unified timbre-aware vocoder $h_\phi$ on both human and non-human audio.

$$f_\theta : (y_{\text{src}}, e_{\text{tgt}}) \mapsto \hat{y}, \qquad (3)$$

subject to

$$\mathcal{M}(\hat{y}) \approx \mathcal{M}(y_{\text{src}}), \mathcal{T}(\hat{y}) \approx e_{\text{tgt}}, \mathcal{T}(\hat{y}) \notin \mathcal{H}. \qquad (4)$$

These formulations naturally extend zero-shot singing voice synthesis (SVS) and singing voice conversion (SVC). When $e_{\text{tgt}} \in \mathcal{H}$, the problem reduces to conventional zero-shot SVS and SVC, where models are trained on parallel singing data $y$ paired with its annotation $S$ or a source singing $y_{\text{src}}$ and its timbre embedding $e$, and inference can be performed using unseen timbre embeddings $e_{\text{tgt}}$.

When the target timbre lies outside the human manifold ($e_{\text{tgt}} \notin \mathcal{H}$), the challenges differ for synthesis and conversion. In NHSVS, the problem is conceptually well-defined but not directly trainable, as non-human audio recordings lack a natural phonetic counterpart and thus cannot be reliably aligned with symbolic musical information $S$, including lyrics or phonemes, for supervision. In contrast, NHSVC remains feasible under a non-parallel training setup through self-reconstruction on human and non-human audio. However, the central challenge remains compared with conventional SVC, where timbre and phoneme information reside within the human vocal domain. In NHSVC, content representations must be carefully designed to capture non-human sounds while remaining compatible with human singing. Moreover, the substantial gap between human and non-human domains in content and timbre requires appropriate training strategies to stabilize optimization during training and reliable performance during inference.

In this way, NHSVS and NHSVC generalize the zero-shot SVS and SVC paradigm to timbre spaces beyond the human distribution, while introducing the central challenge of learning without explicit vocal score alignment in NHSVS, .

### 3.2 CARTOONSING: FRAMEWORK AND MODEL FORMULATION

To address the lack of natural vocal score annotations aligned to non-human audio, we adopt a two-stage modeling strategy based on an intermediate frame-level representation. This formulation allows reliable supervision from annotated human singing and aligned vocal scores while ensuring generalization to both human and non-human timbres during audio synthesis.

#### 3.2.1 UNIFIED REPRESENTATION FOR HUMAN AND NON-HUMAN AUDIO

Conventional SVS systems rely on phoneme- or note-level annotations that provide explicit alignment between symbolic scores and audio. Such annotations, however, are not applicable to non-human audio, as these sounds do not possess a phonetic structure comparable to human singing.

Since direct alignment $\mathcal{M}(\cdot)$ between the symbolic score $S$ and the audio waveform $y$ is infeasible for general non-human audio, we introduce frame-level representations that capture the content and pitch of both human singing and non-human audio.

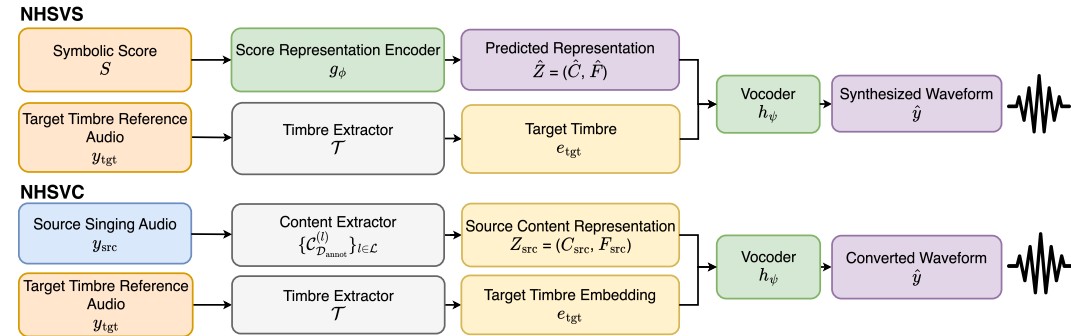

Figure 3: The inference flow of CartoonSing, demonstrating its application in (a) Non-Human Singing Voice Synthesis (NHSVS) from a musical score, and (b) Non-Human Singing Voice Conversion (NHSVC) from a source audio.

Formally, given a recording $y$, either human or non-human, we obtain

$$Z = \Phi(y), \tag{5}$$

where $\Phi$ denotes a general feature extractor and $Z$ is the frame-level representation of $y$ with $T$ frames, containing musical information such as content and pitch.

Specifically, in CartoonSing, we decompose $Z$ into two components:

$$Z = (C, F), \tag{6}$$

where

$$C = \{\mathcal{C}^{(l)}_{\mathcal{D}_{\text{annot}}}(y)\}_{l \in \mathcal{L}}, \quad F = \mathcal{F}_{\text{F0}}(y). \tag{7}$$

Here, $C = \{C^{(l)}\}_{l \in \mathcal{L}}$ is a multi-level sequence of discrete content tokens extracted from a set of selected layers $\mathcal{L}$, and $F = (f_1, \ldots, f_T)$ is a continuous frame-level $F0$ trajectory, where $T$ denotes the number of frames.

Each level of content sequence

$$C^{(l)} = (c^{(l)}_1, \ldots, c^{(l)}_T) \tag{8}$$

is obtained by quantizing timbre-disentangled self-supervised learning (SSL) features extracted at layer $l$ using a K-means model fitted on a subset of annotated human singing data $\mathcal{D}_{\text{annot}} \subset \mathcal{D}_{\text{human}}$. Each example $y \in \mathcal{D}_{\text{annot}}$, which has a corresponding vocal score annotation $S$, is used to construct the token space. The remaining subset $\mathcal{D}_{\text{human}} \setminus \mathcal{D}_{\text{annot}}$, together with non-human recordings $\mathcal{D}_{\text{non-human}}$, is directly quantized to the nearest cluster centroid using the learned K-means codebook. The resulting tokens satisfy

$$c^{(l)}_t \in \{1, \ldots, K^{(l)}\}, \tag{9}$$

where $K^{(l)}$ denotes the number of clusters used for K-means quantization at layer $l$.

This procedure yields multi-layer timbre-disentangled tokens that ensure cross-domain consistency with minimal content loss while suppressing timbre leakage.

For the pitch component $F$, frame-level $F0$ trajectories are estimated directly from the audio using a robust pitch extraction algorithm $\mathcal{F}_{\text{F0}}(\cdot)$. The frame rate of the extracted sequence $F0$ is set to match that of the content token sequence, without explicit alignment with the vocal score.

This factorization is motivated by the symbolic structure of vocal scores, which provide separable supervisory signals corresponding to linguistic content and melodic contour. By explicitly separating content and pitch, the intermediate representations remain interpretable, largely timbre-invariant, and suitable for both human and non-human audio.

The combined representation $Z = (C, F)$ serves as a unified representation for human and non-human recordings and is subsequently used as a supervisory signal in our two-stage synthesis pipeline (Section 3.3).

### 3.3 Two-Stage Formulation of $f_\theta$ for NHSVS and NHSVC

Using the intermediate representation $Z$ defined in Section 3.2.1, we decompose non-human singing generation $f_\theta$ into a two-stage pipeline.

**Stage 1** is a score representation encoder $g_\phi$ that maps a symbolic score $S$ to a frame-level representation $\hat{Z}$:

$$g_\phi : S \mapsto \hat{Z}, \quad \hat{Z} \approx Z. \tag{10}$$

This stage is trained on annotated human singing $\mathcal{D}_{\text{annot}}$ with aligned score annotations $S$.

**Stage 2** is a timbre-conditioned vocoder $h_\psi$ that generates the waveform from a frame-level representation and a timbre reference:

$$h_\psi : (Z, e_{\text{tgt}}) \mapsto \hat{y} \quad \text{s.t.} \quad \Phi(\hat{y}) \approx Z, \ \mathcal{T}(\hat{y}) \approx e_{\text{tgt}}. \tag{11}$$

Unlike Eq. 1, this formulation replaces $S$ with $Z$, so non-human recordings can be used without requiring aligned vocal scores $S$. As a result, $h_\psi$ can be trained on both human and non-human audio, extending the system to non-human timbres.

At inference time, the two tasks are formally defined as

$$\text{NHSVS:} \quad \hat{y} = h_\psi(g_\phi(S), e_{\text{tgt}}), \tag{12}$$

$$\text{NHSVC:} \quad \hat{y}_{\text{tgt}} = h_\psi(\Phi(y_{\text{src}}), e_{\text{tgt}}). \tag{13}$$

Since $h_\psi$ is trained with the constraints in Equations 10 and 11, conditioning on a non-human timbre embedding $e_{\text{tgt}}$ naturally leads to synthesized outputs $\hat{y}$ whose timbre lies outside the human manifold $\mathcal{H}$, i.e., $\mathcal{T}(\hat{y}) \notin \mathcal{H}$.

## 4 Experiments

### 4.1 Experimental Setup

**Datasets** We use 22 Chinese and Japanese open-source singing voice datasets together with 10 non-human audio sources for model training, with full dataset descriptions provided in Appendix A.1. A subset of 13 singing voice datasets $\mathcal{D}_{\text{annot}}$, which contain vocal scores and text annotations, is standardized for phoneme and score alignment and used to train the **Stage-1** score representation encoder $g_\phi$ in Equation 10. For **Stage-2** vocoder $h_\psi$ in Equation 11, all human singing voice and non-human datasets are used for the pretraining, while fine-tuning excludes singing datasets with limited phonetic diversity as well as noisy non-human sources. For domain-specific adaptation, the vocoder is fine-tuned jointly on human singing and one non-human domain at a time, with non-human audio oversampled to achieve a ratio of approximately 0.8:1 to 1:1 relative to human singing. For ablation studies, we additionally train vocoders on a 10% subset of the pretraining data to examine different timbre embedding and content token configurations, with detailed setups provided in Appendix B.1.

**Data Processing** All recordings with an original sampling rate of at least 44.1 kHz are retained, and those above this rate are downsampled to 44.1 kHz for consistency. For $\mathcal{D}_{\text{annot}}$, Japanese lyrics are aligned using `pyopenjtalk` and Chinese phonemes using `ACE_phonemes`. Datasets with score annotations are segmented accordingly, while all other recordings are segmented automatically using energy-based silence detection with recursive refinement, discarding clips longer than 30 seconds. Segments without valid fundamental frequency (F0) estimates are removed from training and evaluation. Detailed descriptions are provided in Appendix A.1.

**Representations** For content representation, we use features from layers $\mathcal{L} = \{5, 8, 9, 12\}$ of the timbre-disentangled self-supervised model ContentVec (Qian et al., 2022), with each layer quantized via K-means clustering ($K = 1024$) to form multi-layered content tokens $C$ in Equation 7. This multi-layer design improves reconstruction ability under our framework constraints. For timbre representation $e_{\text{tgt}}$, we extract embeddings using a pretrained RawNet3 model (Jung et al., 2024a). For pitch representation $F$, we estimate F0 with DIO (Morise et al., 2009) for human singing and CREPE (Kim et al., 2018) for non-singing datasets. Selection considerations are detailed in Appendix A.3. Alternative feature configurations are evaluated in the ablation study (Appendix B.2).

**Models** For **Stage 1**, we adopt a token-based singing voice synthesis acoustic model adapted from XiaoiceSing (Lu et al., 2020) as implemented in (Chang et al., 2024), trained to predict frame-level content tokens and pitch. For **Stage 2**, we adapt the BigVGAN-v2 architecture (Lee et al., 2022) to synthesize waveforms from frame-level F0 trajectories, content token embeddings, and timbre embeddings. Modifications to upsampling ratios and kernel configurations were applied to accommodate our target resolution.

**Domain-Specific Finetuning**

After pretraining, we perform domain-specific finetuning of the **Stage 2** vocoder to improve audio generation quality in settings where non-human timbres are combined with human song references, as in the inference scenario. In these settings, the model is trained to maintain pitch accuracy and intelligibility while accurately reproducing the target timbre (Equation 11).

To achieve this, we extend standard paired training with an *unpaired timbre conditioning* approach. For each training sample in a batch, the source content tokens $C$ and F0 sequence $F$ are randomly paired with a target timbre embedding $e_{\text{tgt}}$. The model first generates a waveform conditioned on $C$, $F$, and $e_{\text{tgt}}$, which is then passed through a shared predictor network to obtain predicted representations $\hat{C}_t^{(l)}$, $\hat{f}_t$, and $\hat{e}_{\text{pred}}$, corresponding to the original content tokens, F0, and target timbre, respectively.

These predictions are used to compute three auxiliary representation prediction losses:

$$\mathcal{L}_{\text{token}} = \frac{1}{T} \sum_{t=1}^{T} \sum_{l \in \mathcal{L}} \text{CE}\big(C_t^{(l)}, \hat{C}_t^{(l)}\big), \tag{14}$$

$$\mathcal{L}_{\text{F0}} = \frac{1}{T} \sum_{t=1}^{T} \text{MSE}\big(f_t, \hat{f}_t\big), \tag{15}$$

$$\mathcal{L}_{\text{timbre}} = 1 - \cos\big(e_{\text{tgt}}, \hat{e}_{\text{pred}}\big), \tag{16}$$

All predictions are produced by a shared predictor network with separate prediction heads for each task. Detailed architecture of the predictor network is provided in Appendix A.5. This design allows the model to utilize supervision from unpaired data and improves the robustness of finetuning.

Finetuning is conducted separately for each domain, including instrumental sounds, bird vocalizations, and general audio. Longer audio segments are used, and training extends beyond paired reconstruction by incorporating unpaired timbre conditioning, thereby enhancing generalization.

**Baselines** For SVS, we use a multi-singer VISinger 2 (Zhang et al., 2023a) model trained on the **Stage 1** datasets $\mathcal{D}_{\text{annot}}$ with standardized vocal score annotations. The model is conditioned on speaker embeddings to support zero-shot inference. For SVC, we adopt Samoye (Wang et al., 2024), a multilingual SVC system, as the only prior work in the singing voice literature that reports evaluations on non-human timbres.

**Evaluation** Our evaluation focuses on non-human singing voice synthesis (NHSVS) and conversion (NHSVC), with an emphasis on timbre similarity, pitch accuracy, and preservation of temporal structure. We consider human references in Chinese and Japanese and non-human timbres, including instrumental, bird, and general sounds, with a primary focus on the instrumental domain due to its cleaner recordings. Each human singing reference is randomly assigned with a target non-human timbre embedding from the corresponding domains. As a comparative baseline, we evaluate the human singing reconstruction.

We evaluate performance using both objective and subjective metrics. Objective metrics include root mean squared error of fundamental frequency (F0 RMSE), voiced/unvoiced error rate (VUV), and timbre similarity (SIM) computed as the cosine similarity between embeddings of the generated audio and the target timbre embedding. We report two variants of timbre similarity: SIM-A, computed using audio embeddings extracted with VGGish (Hershey et al., 2017), and SIM-S, computed using speaker embeddings extracted with RawNet3 (Jung et al., 2024b). Subjective evaluation is conducted with human raters using mean opinion scores for timbre similarity (MOS-T) on Chi-

Table 1: Evaluation on singing voice synthesis and conversion with instrumental timbre for Chinese and Japanese song references. MOS-T is additionally reported for Chinese references.

| Model | Chinese - Instrumental | | | | Japanese - Instrumental | | |
|---|---|---|---|---|---|---|---|
| | LF0 RMSE (↓) | VUV (%) (↓) | SIM-A (↑) | MOS-T (↑) | LF0 RMSE (↓) | VUV (%) (↓) | SIM-A (↑) |
| *SVS* | | | | | | | |
| VISinger 2 | **0.134** | 5.23 | 0.493 | 2.06 | **0.117** | **2.32** | 0.475 |
| CartoonSing (Pretrain) | 0.389 | 5.57 | 0.589 | 3.02 | 0.214 | 2.48 | 0.585 |
| CartoonSing (Finetune) | 0.172 | **5.27** | **0.603** | **3.07** | 0.138 | 2.49 | **0.603** |
| *SVC* | | | | | | | |
| SaMoye-SVC | **0.135** | 5.12 | 0.398 | 2.71 | **0.109** | 2.18 | 0.460 |
| CartoonSing (Pretrain) | 0.254 | 5.44 | 0.570 | 3.01 | 0.171 | 2.31 | 0.548 |
| CartoonSing (Finetune) | 0.147 | **5.04** | **0.589** | **3.20** | 0.114 | **2.10** | **0.576** |

Table 2: Evaluation on singing voice synthesis and conversion with general audio timbre and human song reference.

| Model | Chinese - General | | | Japanese - General | | |
|---|---|---|---|---|---|---|
| | LF0 RMSE (↓) | VUV (%) (↓) | SIM-A (↑) | LF0 RMSE (↓) | VUV (%) (↓) | SIM-A (↑) |
| *SVS* | | | | | | |
| VISinger 2 | **0.168** | 7.954 | 0.441 | **0.119** | 2.67 | 0.435 |
| CartoonSing (Pretrain) | 0.434 | 6.899 | 0.526 | 0.273 | 4.13 | 0.493 |
| CartoonSing (Finetune) | 0.180 | **5.310** | **0.527** | 0.144 | **2.806** | **0.497** |
| *SVC* | | | | | | |
| SaMoye-SVC | **0.142** | 5.07 | 0.461 | **0.112** | **1.97** | 0.450 |
| CartoonSing (Pretrain) | 0.292 | 6.300 | 0.517 | 0.202 | 3.509 | 0.480 |
| CartoonSing (Finetune) | 0.148 | **4.713** | **0.520** | 0.125 | 2.511 | **0.487** |

nese singing generation with instrumental timbre[1]. Detailed dataset splits, metric definitions, and evaluation protocols are provided in Appendix A.7.

## 4.2 MAIN RESULTS

Tables 1, 2, and 3 show that our system consistently demonstrate substantially better similarity with the non-human instrumental timbres than conventional SVS and SVC systems that trained exclusively on human voices.

Comparing pretrained models with domain-specific finetuning, we find that our proposed finetuning strategy yields more stable outputs, reflected in higher MOS-Q scores, and better adherence to musical structure, with improved pitch accuracy and duration consistency. These findings suggest that domain-specific adaptation further enhances generation quality.

Finally, Table 4 presents results on human singing voice reconstruction. Because **Stage 2** training incorporates large amounts of human and non-human audio without requiring vocal score annotations $S$, the proposed framework attains high timbre similarity for human singing. These results indicate that the approach extends to non-human voice generation without introducing degradation in the synthesis quality of human voices.

Audio samples are available at `https://cartoonsing.github.io/`, providing qualitative reference for the reported objective and subjective results.

## 5 CONCLUSION

In this work, we formalize the tasks of Non-Human Singing Voice Synthesis (NHSVS) and Non-Human Singing Voice Conversion (NHSVC), extending the scope of conventional singing voice synthesis and conversion to timbres beyond the human voice. We propose CartoonSing, a unified framework that addresses both tasks in a two-stage synthesis pipeline, enabling zero-shot generation and conversion of singing voices with non-human timbre characteristics.

---

[1]We also evaluate pronunciation clarity (MOS-C) and audio quality (MOS-Q), with results and analysis provided in Appendix A.7.

Table 3: Evaluation on singing voice synthesis and conversion with bird vocalization timbre and human song reference.

| Model | Chinese - Bird | | | Japanese - Bird | | |
|---|---|---|---|---|---|---|
| | LF0 RMSE (↓) | VUV (%) (↓) | SIM-A (↑) | LF0 RMSE (↓) | VUV (%) (↓) | SIM-A (↑) |
| *SVS* | | | | | | |
| VISinger 2 | **0.178** | 7.907 | 0.401 | **0.118** | **2.374** | 0.411 |
| CartoonSing (Pretrain) | 0.448 | 7.517 | 0.446 | 0.296 | 4.667 | 0.427 |
| CartoonSing (Finetune) | 0.190 | **5.312** | **0.492** | 0.152 | 2.986 | **0.453** |
| *SVC* | | | | | | |
| SaMoye-SVC | 0.163 | **5.11** | 0.398 | **0.124** | **2.01** | 0.413 |
| CartoonSing (Pretrain) | 0.302 | 6.903 | 0.437 | 0.234 | 4.358 | 0.419 |
| CartoonSing (Finetune) | **0.157** | 5.143 | **0.471** | 0.125 | 2.658 | **0.440** |

Table 4: Evaluation on human singing voice synthesis and reconstruction.

| Model | Chinese | | | | Japanese | | | |
|---|---|---|---|---|---|---|---|---|
| | LF0 RMSE (↓) | VUV (%) (↓) | SIM-S (↑) | SingMOS (↑) | LF0 RMSE (↓) | VUV (%) (↓) | SIM-S (↑) | SingMOS (↑) |
| VISinger 2 | **0.145** | 7.58 | 0.644 | 2.95 | **0.108** | 2.59 | 0.501 | 2.88 |
| CartoonSing (Pretrain) | 0.310 | 6.88 | 0.661 | 3.05 | 0.190 | 3.02 | 0.525 | 2.94 |
| CartoonSing Vocoder (Pretrain) | 0.230 | **6.33** | **0.683** | **3.12** | 0.141 | **2.39** | **0.578** | **2.98** |

Through comprehensive experiments, we demonstrate that our approach achieves effective timbre transfer for non-human sounds while maintaining synthesis quality for human voices. Domain-specific finetuning further improves generation stability, pitch accuracy, and duration consistency. Moreover, our results show that the proposed framework supports non-parallel training and generalizes across diverse timbre domains without relying on explicit vocal score alignment for non-human audio.

These findings establish a practical and scalable methodology for cross-domain singing voice generation, paving the way for future research in timbre generalization and creative audio synthesis beyond the human voice range.

## 6    ETHICS STATEMENT

All datasets used in this work are publicly available and used according to their respective licensing terms. Human annotations were conducted by listeners under informed consent approved by the IRB[2], with no personally identifiable information included. While our study focuses on non-human timbre singing voice synthesis and conversion for academic purposes, we acknowledge that the methods could also be applied to human voice generation, raising potential risks of unauthorized imitation or copyright infringement. To mitigate such risks, our models will be released with restrictions preventing commercial or unethical use, and future work may incorporate techniques such as vocal watermarking to enhance traceability and safeguard against misuse.

## 7    REPRODUCIBILITY STATEMENT

To support reproducibility, we will release the source code, training scripts, and model configurations upon paper acceptance. All datasets used in our experiments are publicly available and comply with their licensing terms. Detailed training hyperparameters, including optimizer settings, learning rates, warmup steps, and total training steps, are provided in Appendix A.6. Our experiments can be reproduced using a single GPU or a small number of GPUs, and evaluation scripts with exact metric computation will also be made available. Hardware specifications are documented to ensure that the reported results can be faithfully reproduced.

---

[2]The IRB protocol number will be provided after paper acceptance to preserve anonymity

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

## A    DETAILED SETUP FOR MAIN EXPERIMENTS

### A.1    DATASETS

Table 5 lists all human singing voice datasets and their use in our experiments. Table 6 lists all non-human datasets and their use in our experiments.

### A.2    DATA PROCESSING

All audio was processed at a 44.1 kHz sampling rate. For the 13 datasets $\mathcal{D}_{\mathrm{annot}}$ used in **Stage 1** training, Japanese lyrics annotations were aligned using `pyopenjtalk`[3], and Chinese phoneme annotations were aligned using `ACE_phonemes`[4]. We segmented 20 singing datasets based on their vocal score annotations. These datasets include: ACE-KiSing (Shi et al., 2024), ACE-Opencpop (Shi et al., 2024), Amaboshi CipherDB2 (Chikano, 2024), Itako (SSS LLC, 2021),

---

[3] https://github.com/r9y9/pyopenjtalk
[4] https://github.com/timedomain-tech/ACE_phonemes

Table 5: Singing voice datasets used in our experiments. "✓" indicates dataset usage in the corresponding training stage; blank cells indicate not used.

| Dataset | Lang (Used) | Stage 2 Pretrain | Stage 2 Fine-tune | Stage 1 |
|---|---|---|---|---|
| ACE-KiSing (Shi et al., 2024) | zh | ✓ | ✓ | ✓ |
| ACE-Opencpop (Shi et al., 2024) | zh | ✓ | ✓ | ✓ |
| Amaboshi CipherDB2 (Chikano, 2024) | jp | ✓ | ✓ | ✓ |
| GTSinger (Zhang et al., 2024c) | zh/jp | ✓ | ✓ | |
| Itako (SSS LLC, 2021) | jp | ✓ | ✓ | ✓ |
| JaCappella (Nakamura et al., 2023) | jp | ✓ | | |
| JSUT Song Corpus (Takamichi et al., 2017) | jp | ✓ | ✓ | |
| Kiritan (Ogawa & Morise, 2021) | jp | ✓ | ✓ | ✓ |
| KiSing (Shi et al., 2022) | zh | ✓ | ✓ | ✓ |
| M4Singer (Zhang et al., 2022) | zh | ✓ | ✓ | ✓ |
| Namine Ritsu (Canon, 2009) | jp | ✓ | ✓ | ✓ |
| Natsume Yuuri (Kei & Sota, 2020) | jp | ✓ | ✓ | ✓ |
| NIT SONG070 F001 (HTS Working Group, 2015) | jp | ✓ | ✓ | |
| No.7 Singing Database (Morise et al., 2022) | jp | ✓ | ✓ | |
| OfutonP (OfutonP, 2020) | jp | ✓ | ✓ | ✓ |
| Onikuru Kurumi (Kurumi, 2020) | jp | ✓ | ✓ | ✓ |
| Opencpop (Wang et al., 2022) | zh | ✓ | ✓ | ✓ |
| OpenSinger (Huang et al., 2021) | zh | ✓ | ✓ | |
| PJS (Koguchi et al., 2020) | jp | ✓ | ✓ | ✓ |
| PopCS (Liu et al., 2022) | zh | ✓ | ✓ | |
| SingStyle111 (Dai et al., 2023) | zh | ✓ | ✓ | |
| VocalSet (Wilkins et al., 2018) | vowels | ✓ | ✓ | |

Table 6: Non-vocal datasets used for vocoder pretraining and fine-tuning. "✓" indicates dataset usage in the corresponding stage; blank cells indicate not used.

| Dataset | Category | Stage 2 Pretrain | Stage 2 Fine-tune |
|---|---|---|---|
| BBC Audio Effects (BBC) | General | ✓ | |
| CCOM-HuQin (Zhang et al., 2023b) | Instrumental | ✓ | ✓ |
| Cornell Birdcall Identification (Howard et al., 2020) | Bird Vocalization | ✓ | ✓ |
| FSD50K (Fonseca et al., 2021) | General | ✓ | ✓ |
| GameAudioGDC (2015–2024) (SONNISS, 2024) | General | ✓ | |
| GoodSounds (Romani et al., 2015) | Instrumental | ✓ | ✓ |
| JaCappella (Nakamura et al., 2023) | Vocal Percussion | ✓ | ✓ |
| MUSDB18-HQ (Rafii et al., 2019) | Instrumental | ✓ | ✓ |
| Philharmonia Sound Samples (Philharmonia) | Instrumental | ✓ | ✓ |
| URMP (Ouyang et al., 2023) | Instrumental | ✓ | ✓ |

JSUT Song Corpus (Takamichi et al., 2017), Kiritan (Ogawa & Morise, 2021), M4Singer (Zhang et al., 2022), Namine Ritsu (Canon, 2009), Natsume Yuuri (Kei & Sota, 2020), OfutonP (OfutonP, 2020), NIT SONG070 F001 (HTS Working Group, 2015), Onikuru Kurumi (Kurumi, 2020), and Opencpop (Wang et al., 2022).

For all other singing and non-vocal audio, segmentation was performed automatically. We applied energy-based silence detection using SoX to recursively divide long recordings into utterances. Silence detection was first performed on the raw audio, and segments longer than 15 seconds were re-segmented with progressively adjusted parameters for up to three iterations. Only clips of at most 30 seconds were retained, and longer segments were discarded.

Additionally, segments without valid fundamental frequency (F0) estimates were excluded from training. In early experiments, we did not apply such F0 filtering and included all audio for training. However, we observed degraded pitch control in the generated output. Further analysis revealed that certain pitch extraction methods failed to produce valid F0 predictions for some non-human audio, introducing noise into training. To ensure training stability, segments with all-zero F0 values were filtered out. This experience also guided the selection of a suitable pitch extractor for non-human audio, as detailed in Appendix A.3.

## A.3 REPRESENTATION SELECTION DETAILS AND CONSIDERATIONS

For content representation, we use features from layers $\mathcal{L} = \{5, 8, 9, 12\}$ of timbre-disentangled self-supervised model ContentVec (Qian et al., 2022) to form the multi-layered content tokens $C$ in Equation 7, with feature vectors from each layer quantized via K-means clustering ($K = 1024$). These layers were chosen as follows: layer 5 is the earliest layer where contrastive loss is applied for feature disentanglement; layer 8 was reported to yield the best voice conversion performance in the original paper; layer 9 exhibits the lowest speaker information content according to their speaker identification (SID) accuracy figure; and layer 12 corresponds to the final representation layer. The multi-layer setup is used to improve audio reconstruction ability, given the reduced acoustic information in our framework setting. Alternative content representations are investigated in the ablation study (Section B.2).

For pitch representation $F$, specifically the fundamental frequency (F0), we use DIO (Morise et al., 2009) for human singing recordings and CREPE (Kim et al., 2018) for non-singing datasets. We compared several mainstream F0 extraction methods, including DIO (Morise et al., 2009), CREPE (Kim et al., 2018), Harvest (Morise et al., 2017), Parselmouth (Jadoul et al., 2018), YIN (De Cheveigné & Kawahara, 2002), and pYIN (Mauch & Dixon, 2014), on a subset of non-human datasets to assess their reliability. We computed the proportion of segments having valid F0 predictions, along with the mean and range of predicted F0 values, and performed small-scale qualitative listening tests for verification. CREPE was selected for its ability to produce valid predictions for nearly all segments while maintaining high perceived pitch accuracy.

Unless otherwise specified, all downstream models consume the features described above.

## A.4 MODEL DETAILS

### A.4.1 SCORE REPRESENTATION ENCODER

The score representation encoder $g_\theta$ is a non-autoregressive, XiaoiceSing-style (Lu et al., 2020) Transformer (Ren et al., 2021) that maps a symbolic score $S$ to a frame-level acoustic representation $\hat{Z}$. We denote the score as a sequence of tuples $S = \{(p_i, n_i, d_i)\}_{i=1}^N$, where $p_i$ is the phoneme, $n_i$ is the MIDI note number, $d_i$ is the ground-truth phoneme duration in frames, and $N$ is the total number of phonemes in the score.

**Encoder Architecture.** The encoder is a 6-layer Transformer that uses relative self-attention and 1D convolutions in its position-wise feed-forward networks, similar to the Conformer architecture. It is configured with an attention dimension of 384 and 2 attention heads .

**Duration and Pitch Predictors.** Following the encoder, two separate predictor modules estimate prosodic features:

- **Duration Predictor:** A simple feed-forward network predicts the log-duration of each input phoneme.
- **Pitch Predictor:** Another feed-forward network predicts the frame-level log-F0 contour from the length-regulated hidden states.

During training, we use ground-truth durations from forced alignment to expand the encoder states to the frame-level via a length regulator. At inference, the predicted durations are used instead.

**Decoder and Projection.** The final frame-level hidden states, conditioned on the predicted pitch, are processed by a 6-layer Transformer decoder. The decoder's output is passed through a linear projection layer to produce a sequence of continuous feature vectors, forming the intermediate representation $\hat{Z}$.

**Training Objectives.** The model is trained end-to-end using a weighted multi-task objective, $\mathcal{L}$, defined as:

$$\mathcal{L} = \lambda_{\text{out}}\mathcal{L}_{\text{out}} + \lambda_{\text{dur}}\mathcal{L}_{\text{dur}} + \lambda_{\text{pitch}}\mathcal{L}_{\text{pitch}},$$

where $\lambda_{\text{out}} = \lambda_{\text{dur}} = \lambda_{\text{pitch}} = 1.$ are the loss weights for each component. The individual loss components are:

- **Output Loss ($\mathcal{L}_{\textbf{out}}$):** The L1 loss between the predicted logits and the target discrete tokens.
- **Duration Loss ($\mathcal{L}_{\textbf{dur}}$):** The L1 loss between the predicted and ground-truth log-durations.
- **Pitch Loss ($\mathcal{L}_{\textbf{pitch}}$):** The L1 loss between the predicted and ground-truth log-F0, computed only on voiced frames.

### A.4.2 UNIFIED TIMBRE-AWARE VOCODER

The second stage of our framework is a unified timbre-aware vocoder, $h_\phi$, that synthesizes the final waveform $\hat{y}$ from the frame-level representation $Z = (C, F)$ (composed of content tokens $C$ and a pitch contour $F$) and a target timbre embedding $e_{\text{tgt}}$. Our implementation adapts the BigVGAN-v2 architecture (Lee et al., 2022) by modifying its input conditioning to accept multi-layer discrete content tokens, F0, and timbre embeddings. The upsampling ratios are adjusted to accommodate our feature frame rate. This design enables joint adversarial training on both human and non-human audio, allowing the model to generalize across a wide range of timbres.

**Input Conditioning.** The vocoder is designed to handle the diverse inputs required for both human and non-human synthesis:

- **Content Tokens ($C$):** The input content is represented by four layers of discrete tokens. Each layer has its own embedding table. The resulting embeddings are combined into a single representation using a learned weighted sum.
- **Pitch ($F$):** The frame-level log-F0 sequence is passed through a linear layer to create a pitch embedding, which is then concatenated with the content embedding.
- **Timbre ($e_{\text{tgt}}$):** The target timbre embedding is projected by a fully-connected layer and added to the combined content and pitch representation. This allows the model to adapt to arbitrary target timbres in a zero-shot manner.

**Generator Architecture.** The generator is a fully convolutional model that upsamples the conditional input features from a 20ms frame rate to the target audio sampling rate (44.1 kHz). It consists of a series of transposed convolutional layers, resulting in a total upsampling factor of 882, which matches the hop size. Between each upsampling layer, a stack of Anti-aliased Multi-Periodicity (AMP) residual blocks with kernel sizes $[3, 7, 11]$ and dilations of $[1, 3, 5]$ process the features. We use the 'snakebeta' periodic activation function (Lee et al., 2022) to effectively model the periodic nature of audio signals.

**Adversarial Training.** The vocoder is trained adversarially against a multi-resolution, multi-period discriminator. The discriminator architecture combines a Multi-Scale Sub-Band Constant-Q Transform (CQT) Discriminator (Gu et al., 2024) and a Multi-Period Discriminator (MPD) from HiFi-GAN (Kong et al., 2020). The training objective is a combination of a GAN loss and a feature matching loss, with weights $\lambda_{\text{adv}} = 1.0$ and $\lambda_{\text{feat\_match}} = 2.0$, respectively. We also incorporate a multi-scale mel-spectrogram reconstruction loss with a weight of $\lambda_{\text{aux}} = 15.0$ to further improve generation quality. The model is trained jointly on both the human singing datasets and the non-human audio datasets.

### A.5 PREDICTOR NETWORK ARCHITECTURE

The predictor network employed for auxiliary representation estimation is implemented as a stack of one-dimensional convolutional layers followed by task-specific linear projection heads. The network processes generated audio waveforms to produce predictions for content tokens, F0, and timbre embeddings.

### A.5.1 CONVOLUTIONAL STACK

The convolutional backbone consists of 7 layers with the following parameters:

- Input channels: 1
- Kernel sizes: $[10, 3, 3, 3, 3, 2, 2]$
- Strides: $[7, 7, 3, 3, 2, 1, 1]$
- Paddings: $[4, 1, 1, 1, 1, 1, 1]$
- Hidden channels: $[512, 512, 512, 512, 512, 512, 512]$
- Bias: applied to all convolutional layers
- Activation function: LeakyReLU with negative slope $0.1$
- Weight normalization: applied to all convolutional layers

Let $x^{(0)}$ denote the input to the predictor. The $l$-th convolutional layer is formally defined as

$$x^{(l)} = \phi\Big(\text{Conv1d}\big(x^{(l-1)}, k_l, s_l, p_l, b_l\big)\Big),$$

where $\phi$ denotes the LeakyReLU activation, $k_l$, $s_l$, $p_l$ and $b_l$ correspond to the kernel size, stride, padding, and bias of layer $l$.

### A.5.2 PROJECTION HEADS

For each prediction target, a dedicated linear projection maps the final hidden representation to the target dimension. Let $\mathbf{h}^{(7)}$ denote the output of the last convolutional layer. The predicted representations are computed as

$$\hat{y}_i = \text{Linear}_i(\mathbf{h}^{(7)}), \quad i \in \{\text{f0}, \text{token}, \text{spemb}\}.$$

The target dimensions are specified as follows:

- **F0:** $[1]$
- **Token:** $[1025, 4]$
- **Speaker embedding (spemb):** $[192]$

This architecture allows the predictor to share a common hidden representation while producing multiple auxiliary outputs, supporting the token classification, F0 regression, and timbre embedding prediction tasks described in Section 4.1. The design balances expressivity and parameter efficiency through deep convolutional feature extraction combined with lightweight linear projections for each task.

### A.6 TRAINING DETAILS

Table 7: Training hyperparameters for score representation encoder in Stage 1.

| Hyperparameter | Value |
|---|---|
| Max Epochs | 70 |
| Batch Size | 16 |
| Gradient Clip Norm | 1.0 |
| Optimizer | Adam |
| Learning Rate | 5.0e-4 |

Table 8: Training schedule for Stage 2 vocoder in main experiments. Each training sample is a randomly cropped fixed-length segment of the indicated size.

| Experiment | Training Steps | Batch Size | Segment Length (samples) |
|---|---|---|---|
| Pretrain | 350,000 | 8 | 16,384 |
| Finetune | 80,000 | 4 | 32,768 |

Table 9: Shared hyperparameters and optimizer settings for Stage 2 vocoder experiments.

| Hyperparameter | Value |
|---|---|
| Warmup Steps | 40,000 |
| Warmup Gradient Clip Norm | 100 |
| Training Gradient Clip Norm | 500 |
| Optimizer | AdamW |
| LR Scheduler | ExponentialLR |
| Learning Rate | 0.0001 |

We train our vocoder models on a single V100 GPU and score representation encoder on two V100 GPUs. The score representation encoder is optimized with Adam with a peak learning rate of $5.0 \times 10^{-4}$, while the vocoder is optimized with AdamW with a peak learning rate of $1.0 \times 10^{-4}$. For vocoder training in the main experiments, we use 40k warmup steps followed by 350k training steps for pretraining, and 40k warmup steps with 60k training steps for fine-tuning (Appendix A.6). For ablation studies, the vocoder is trained with 40k warmup steps and 250k training steps (Appendix B.1).

We summarize our training and optimization settings for both stages. All **Stage 1** score representation encoder experiments use the same configuration, listed in Table 7. **Stage 2** vocoder experiments vary in training steps, batch size, and segment length (Table 8), while other hyperparameters are shared (Tables 9).

### A.7 EVALUATION

#### A.7.1 DATA

We construct our test sets as follows. For Chinese singing voices, evaluations are conducted on the Opencpop (Wang et al., 2022), KiSing (Shi et al., 2022), and multi-singer ACE-KiSing (Shi et al., 2024) test sets. For Japanese singing voices, we used test sets of Amaboshi CipherDB2 (Chikano, 2024), Itako (SSS LLC, 2021), Kiritan (Ogawa & Morise, 2021), Namine Ritsu (Canon, 2009), Natsume Yuuri (Kei & Sota, 2020), OfutonP (OfutonP, 2020), and Onikuru Kurumi (Kurumi, 2020). For non-human datasets, if a dataset does not provide an official test split, we randomly select 10% of segments as the test set. For instrumental evaluation, we exclude noisy subsets such as "others" and "drums" from MUSDB18-HQ (Rafii et al., 2019), while retaining all other instrumental test sets. For general sounds, we use FSD50K as a clean test set. For bird sounds, since the Cornell Birdcall Identification (Howard et al., 2020) dataset only publicly releases three samples in its official test set, as it was designed for a Kaggle competition, we randomly sample 10% of its training set to serve as a test set.

#### A.7.2 OBJECTIVE EVALUATION

Objective metrics include root mean squared error of F0 and voiced/unvoiced error rate, computed using the Harvest F0 estimator (Morise et al., 2017), to measure pitch and timing accuracy. Timbre similarity is computed as the cosine similarity between embeddings of the generated audio and the target timbre. Embeddings are extracted using either VGGish (Hershey et al., 2017) or a pretrained RawNet3 speaker embedding model (Jung et al., 2024b). In the result tables, timbre similarity is reported as SIM-A when computed with audio embeddings using VGGish, and SIM-S when computed with speaker embeddings using RawNet3. For human singing voice reconstruction, we additionally report an automated singing voice MOS prediction metric, SingMOS (Tang et al., 2024).

### A.8 SUBJECTIVE EVALUATION

60 pairs of synthesized samples were randomly selected for subjective evaluation. 13 listeners participated in a blind, randomized listening evaluation on a voluntary basis. Participants provided informed consent before participating and were instructed to evaluate samples independently, without discussion or influence from others, basing their judgments on their own perception.

Table 10: MOS evaluation on singing voice synthesis and conversion with instrumental timbre for Chinese song references.

| Model | MOS-T (↑) | MOS-C (↑) | MOS-Q (↑) |
|---|---|---|---|
| *SVS* | | | |
| VISinger 2 | 2.06 | **3.82** | **3.90** |
| CartoonSing (Pretrain) | 3.02 | 2.76 | 2.44 |
| CartoonSing (Finetune) | **3.07** | 2.99 | 2.75 |
| *SVC* | | | |
| Samoye | 2.71 | **4.09** | **4.02** |
| CartoonSing (Pretrain) | 3.01 | 2.89 | 2.57 |
| CartoonSing (Finetune) | **3.20** | 3.01 | 2.80 |

Listeners were instructed to rate samples along three dimensions: timbre similarity (MOS-T), intelligibility (MOS-C), and audio quality (MOS-Q), each on a Likert scale from 1 (lowest) to 5 (highest). For MOS-T, listeners were instructed to assess the degree to which the synthesized voice matches the target timbre, regardless of differences in other attributes. For MOS-C, listeners focused on pronunciation clarity, independent of timbre or synthesis quality. For MOS-Q, listeners assessed the overall audio quality, specifically the cleanliness of the synthesized audio, independent of timbre similarity and clarity.

Each dimension was rated separately to ensure independent assessment of each aspect of synthesis quality. This procedure was designed to ensure objectivity, consistency, and reproducibility in subjective evaluation.

Subjective evaluation shows that our proposed system CartoonSing achieves substantially higher similarity to instrumental timbre compared to baseline models. However, timbre transfer to non-human voices introduces a trade-off in audio quality and clarity. We find that clarity is primarily affected by the weakening of consonantal articulation when the generated voice more closely matches instrumental timbre. This arises from the acoustic mismatch between speech and instrumental sounds: speech intelligibility depends on transient consonant cues such as plosives and fricatives, whereas instrumental sounds are generally vowel-like, characterized by stable resonances and harmonic structures with limited transient components. Consequently, consonant-like details become less salient under strongly non-human timbres, leading to reduced MOS-C. Existing systems do not exhibit this trade-off because they do not faithfully reproduce instrumental timbre. This finding points to a broader research challenge in non-human speech generation (NHSG) for future research, namely how to better capture and synthesize consonant-like transients in the presence of strongly non-human timbres.

# B  ABLATION STUDY

## B.1  EXPERIMENTAL SETUP

All ablation experiments follow the same configuration as the main experiments unless otherwise noted. To improve computational efficiency, we randomly sample 10% of the human and non-human pretraining datasets while preserving their relative proportions, reduce Stage 2 training to 250k steps, and omit fine-tuning.

Evaluation is conducted on the same datasets as in the main experiments, reporting only objective metrics. In this setup, some outputs cannot be reliably processed by the F0 predictor during the computation of LF0 RMSE. Therefore, we additionally report the failure rates of F0 extraction, denoted as F0 NaN, in the results tables for reference. LF0 RMSE is computed only over segments with valid F0 values.

## B.2  ABLATIONS ON CONTENT REPRESENTATION

We replace the ContentVec tokens used in the main system with alternative content representations. In particular, we compare ContentVec tokens with HuBERT tokens, where the layer selection for HuBERT follows the same setting as for ContentVec. As shown in Tables 11, 12, and 13, HuBERT

Table 11: Ablation study on content and timbre representation choices for singing voice synthesis (SVS) and conversion (SVC) with human → instrumental.

| Model | Chinese - Instrumental | | | | Japanese - Instrumental | | | |
|---|---|---|---|---|---|---|---|---|
| | LF0 RMSE (↓) | F0 NaN (%) (↓) | VUV (%) (↓) | SIM-A (↑) | LF0 RMSE (↓) | F0 NaN (%) (↓) | VUV (%) (↓) | SIM-A (↑) |
| *SVS Ablation* | | | | | | | | |
| CartoonSing-ablation | **0.171** | **0.00** | 6.03 | 0.576 | **0.139** | **0.00** | 3.22 | 0.578 |
| *w/ AudioMAE* | 0.411 | **0.00** | 6.13 | 0.618 | 0.325 | **0.00** | 3.49 | 0.598 |
| *w/ CLAP-Large* | 0.371 | **0.00** | **5.78** | **0.711** | 0.291 | **0.00** | **2.40** | **0.705** |
| *w/ HuBERT* | 0.291 | **0.00** | 6.82 | 0.560 | 0.211 | **0.00** | 3.86 | 0.539 |
| *SVC Ablation* | | | | | | | | |
| CartoonSing-ablation | 0.335 | **0.00** | 5.96 | 0.604 | 0.263 | **0.00** | 3.20 | 0.576 |
| *w/ AudioMAE* | **0.145** | **0.00** | 6.03 | 0.558 | **0.120** | **0.00** | **2.97** | 0.555 |
| *w/ CLAP-Large* | 0.294 | **0.00** | **5.73** | **0.704** | 0.256 | **0.00** | 2.30 | **0.685** |
| *w/ HuBERT* | 0.214 | **0.00** | 6.54 | 0.526 | 0.172 | **0.00** | 3.54 | 0.502 |

Table 12: Ablation study on content and timbre representation choices for singing voice synthesis (SVS) and conversion (SVC) with human → general audio.

| Model | Chinese - General | | | | Japanese - General | | | |
|---|---|---|---|---|---|---|---|---|
| | LF0 RMSE (↓) | F0 NaN (%) (↓) | VUV (%) (↓) | SIM-A (↑) | LF0 RMSE (↓) | F0 NaN (%) (↓) | VUV (%) (↓) | SIM-A (↑) |
| *SVS Ablation* | | | | | | | | |
| CartoonSing-ablation | **0.218** | **0.00** | 9.08 | 0.545 | **0.171** | **0.00** | 7.14 | 0.533 |
| *w/ AudioMAE* | 0.505 | 0.02 | 17.04 | **0.586** | 0.416 | 0.78 | 15.36 | **0.572** |
| *w/ CLAP-Large* | 0.993 | **0.00** | **7.36** | 0.575 | 0.829 | 0.11 | **6.83** | 0.557 |
| *w/ HuBERT* | 0.425 | 0.02 | 15.42 | 0.541 | 0.371 | 0.22 | 13.18 | 0.502 |
| *SVC Ablation* | | | | | | | | |
| CartoonSing-ablation | 0.429 | 0.02 | 24.37 | **0.583** | 0.386 | 0.44 | 15.25 | **0.556** |
| *w/ AudioMAE* | **0.183** | **0.00** | 11.31 | 0.543 | **0.164** | **0.00** | 6.80 | 0.514 |
| *w/ CLAP-Large* | 0.914 | **0.00** | **10.80** | 0.574 | 0.771 | **0.00** | **6.47** | 0.552 |
| *w/ HuBERT* | 0.330 | **0.00** | 14.21 | 0.536 | 0.331 | 0.33 | 13.23 | 0.487 |

achieves higher timbre similarity in human singing reconstruction but performs worse in timbre similarity when synthesizing with non-human timbres. We hypothesize that this is because HuBERT encodes richer acoustic information and does not explicitly disentangle timbre from linguistic content, which may lead to timbre leakage when combined with non-human timbre embeddings. This observation highlights the importance of disentangled content representations in non-human singing synthesis.

### B.3 ABLATIONS ON TIMBRE ENCODER

To examine the effect of timbre representations, we train Stage 2 models with embeddings from AudioMAE (Huang et al., 2022) and CLAP (Wu et al., 2023), in addition to the RawNet3 embeddings used in the main experiments (Jung et al., 2024a). As shown in Tables 11, 12, and 13, no single timbre encoder consistently outperforms the others across all evaluation conditions. CLAP-Large generally achieves high timbre similarity in non-human transfer and reconstruction tasks (including Table 15), but shows comparatively weaker performance in preserving human timbre (Table 14). These findings suggest that the choice of timbre encoder is closely tied to the target domain and task, and that different encoders offer complementary advantages rather than a universally superior solution.

## C THE USE OF LARGE LANGUAGE MODELS

We used large language models (LLMs) to aid in polishing the writing of this paper. Their usage was limited to language refinement, LaTeX formatting, and icon creations in the diagrams, and did not contribute to research ideas, design, implementation, or analysis.

Table 13: Ablation study on content and timbre representation choices for singing voice synthesis (SVS) and conversion (SVC) with human → bird audio.

| Model | Chinese - Bird | | | | Japanese - Bird | | | |
|---|---|---|---|---|---|---|---|---|
| | LF0 RMSE (↓) | F0 NaN (%) (↓) | VUV (%) (↓) | SIM-A (↑) | LF0 RMSE (↓) | F0 NaN (%) (↓) | VUV (%) (↓) | SIM-A (↑) |
| *SVS Ablation* | | | | | | | | |
| CartoonSing-ablation | **0.272** | **0.00** | 10.29 | **0.521** | **0.226** | 0.33 | 9.31 | **0.509** |
| *w/ AudioMAE* | 0.463 | 0.02 | 15.95 | 0.517 | 0.374 | 0.67 | 15.08 | 0.508 |
| *w/ CLAP-Large* | 0.939 | **0.00** | **6.89** | 0.509 | 0.841 | **0.11** | **6.09** | 0.501 |
| *w/ HuBERT* | 0.446 | 0.04 | 17.37 | 0.485 | 0.380 | 0.89 | 14.19 | 0.448 |
| *SVC Ablation* | | | | | | | | |
| CartoonSing-ablation | **0.232** | **0.00** | 12.42 | **0.522** | **0.207** | 0.22 | 9.09 | 0.498 |
| *w/ AudioMAE* | 0.382 | 0.02 | 16.25 | 0.516 | 0.324 | 0.33 | 14.73 | 0.498 |
| *w/ CLAP-Large* | 0.878 | 0.02 | **6.93** | 0.511 | 0.806 | **0.00** | **5.83** | **0.499** |
| *w/ HuBERT* | 0.362 | 0.02 | 17.25 | 0.484 | 0.344 | 0.67 | 14.55 | 0.441 |

Table 14: Evaluation on human singing voice synthesis and reconstruction.

| Model | Chinese - Human | | | | Japanese - Human | | | |
|---|---|---|---|---|---|---|---|---|
| | LF0 RMSE (↓) | VUV (%) (↓) | SIM-S (↑) | SingMOS (↑) | LF0 RMSE (↓) | VUV (%) (↓) | SIM-S (↑) | SingMOS (↑) |
| *SVS Ablation* | | | | | | | | |
| CartoonSing-ablation | **0.170** | 8.12 | 0.664 | **3.096** | **0.127** | **2.91** | 0.515 | 2.906 |
| *w/ AudioMAE* | 0.323 | 7.20 | 0.540 | 2.824 | 0.237 | 3.13 | 0.452 | 2.801 |
| *w/ CLAP-Large* | 0.248 | 7.46 | 0.477 | 2.967 | 0.174 | 3.02 | 0.381 | 2.876 |
| *w/ HuBERT* | 0.330 | **7.08** | **0.661** | 3.048 | 0.200 | 3.06 | **0.524** | **2.923** |
| *Vocoder Ablation* | | | | | | | | |
| CartoonSing-ablation | **0.146** | 4.41 | **0.681** | **3.146** | **0.106** | 2.41 | 0.561 | **2.965** |
| *w/ AudioMAE* | 0.246 | 4.53 | 0.565 | 2.905 | 0.154 | 2.43 | 0.500 | 2.848 |
| *w/ CLAP-Large* | 0.180 | 6.57 | 0.495 | 3.033 | 0.156 | 2.41 | 0.417 | 2.939 |
| *w/ HuBERT* | 0.265 | **4.32** | 0.680 | 3.073 | 0.176 | **2.35** | **0.580** | 2.928 |

Table 15: Evaluation on beyond-human audio reconstruction.

| Model | Instrumental | | General | | Bird | |
|---|---|---|---|---|---|---|
| | MCD (↓) | SIM-A (↑) | MCD (↓) | SIM-A (↑) | MCD (↓) | SIM-A (↑) |
| CartoonSing-ablation | 8.86 | 0.803 | 12.10 | 0.687 | 11.01 | 0.851 |
| *w/ AudioMAE* | 7.61 | **0.822** | **9.49** | **0.713** | **9.03** | **0.867** |
| *w/ CLAP-Large* | **7.34** | 0.804 | 10.13 | 0.687 | 10.01 | 0.700 |
| *w/ HuBERT* | 8.60 | 0.752 | 11.86 | 0.693 | 10.60 | 0.866 |

