# OpenReview forum: "CartoonSing: Unifying Human and Nonhuman Timbres in Singing Generation"
_ICLR.cc/2026/Conference — ICLR 2026 Conference Withdrawn Submission_

### Official Review · Reviewer_9QPj · 2025-10-29

**Soundness:** 3
**Presentation:** 3
**Contribution:** 2
**Rating:** 4
**Confidence:** 3

**Summary:**

This paper introduces CartoonSing, a two-stage pipeline for Non-Human Singing Voice Synthesis (NHSVS). The authors frame the problem as generating singing with non-human timbres (e.g., instruments, birds) while preserving pitch, content, and articulation. The system uses (1) a ContentVec + F0–based acoustic model to produce discrete content tokens and pitch curves from score input, and (2) a BigVGAN-v2-based vocoder conditioned on tokens, F0, and a timbre embedding (RawNet3). The model is pretrained on human singing datasets and then finetuned per non-human domain. The authors evaluate the system using objective similarity metrics and MOS-style perceptual ratings.

**Strengths:**

* The idea of extending singing synthesis beyond human vocal timbres is interesting and underexplored.

* Combining content tokens, pitch estimation, and timbre embeddings is straightforward and simple approach

* Some comparisons between timbre encoders (e.g., RawNet3 vs. CLAP) are included.

**Weaknesses:**

* The system is largely a recombination of existing components (ContentVec tokens, RawNet3 embeddings, BigVGAN-v2 vocoder). Claims of unification are weakened by the need for per-domain finetuning.

* No confidence intervals or statistical tests are reported.

* Baselines (VISinger2, SaMoye) are only trained on human vocals, while CartoonSing is finetuned on non-human data. Improvements in similarity are therefore confounded by data exposure rather than architecture.


* RawNet3 is a speaker verification model; using it to represent instruments/birds is questionable.

* Audio quality (MOS-Q) degrades relative to baselines. This mirrors my impressions of the samples which are not very convincing.

* The samples are very limited and only use intruments.

* Claims of zero-shot generalization are inconsistent with reliance on domain-specific finetuning.

This is an interesting problem framing with a reasonable first-pass system, but the quality is not yet competitive, and the architecture is incremental over existing work.

**Questions:**

* Can you provide MOS/MUSHRA (with CIs) for all  domains?

* Can baselines be trained on the same non-human data for fairness?

---

### Official Review · Reviewer_djaW · 2025-10-30

**Soundness:** 3
**Presentation:** 3
**Contribution:** 2
**Rating:** 2
**Confidence:** 3

**Summary:**

This paper introduces an interesting new problem: Non-Human Singing Generation. To address this, the authors propose a unified framework for both Singing Voice Synthesis (SVS) and Singing Voice Conversion (SVC), along with a two-stage pipeline: (1) a score representation encoder trained on annotated human singing data, and (2) a timbre-aware vocoder that reconstructs waveforms for both human and non-human audio.

**Strengths:**

This paper introduces an interesting new problem: Non-Human Singing Generation. To tackle this, the authors propose a unified framework for Singing Voice Synthesis (SVS) and Singing Voice Conversion (SVC), leveraging self-supervised learning (SSL) features to disentangle timbre, content, and F0, thereby enabling effective timbre transfer.

**Weaknesses:**

The disentanglement approach proposed in this paper has already been widely adopted in voice conversion (VC) and text-to-speech (TTS) systems, which weakens the novelty of the work. As a result, the contribution appears to be a relatively trivial extension of these established methods to the domain of non-human singing.

**Questions:**

Could the capability for non-human singing naturally emerge if large-scale web data and an autoregressive architecture are used?

---

### Official Review · Reviewer_FTZh · 2025-10-31

**Soundness:** 2
**Presentation:** 3
**Contribution:** 1
**Rating:** 2
**Confidence:** 4

**Summary:**

The paper introduces and formally defines NHSVS and NHSVC, and proposes CartoonSing, a two-stage pipeline (score-representation encoder + timbre-aware vocoder) to generate singing with non-human timbres. The method factorizes content/pitch and conditions a vocoder on timbre embeddings.

**Strengths:**

1. A clear two-stage formulation that reduces reliance on non-human aligned data by training the score encoder on human singing only, then learning a unified vocoder over human + non-human audio.
2. The paper is generally well-organized and easy to follow.

**Weaknesses:**

1. The task defined in this paper is rather narrow and lacks clear real-world applications. The definition of “non-human” is also somewhat vague. Although the authors claim that the proposed task could apply to video games, movies, and virtual characters, the presented demos mainly target instrument-like timbres instead of genuinely non-human vocal styles.
2. The overall demo quality is poor—the generated sentences are short, the audio quality is low, and even the lyrics are often unintelligible.
3. The paper compares only with a very limited set of baselines. To properly evaluate timbre generalization and zero-shot performance, it should include several end-to-end timbre-controllable SVS and SVC systems as additional references, such as the following:

 [1]  Zhang, Yu, et al. "TCSinger 2: Customizable Multilingual Zero-shot Singing Voice Synthesis." arXiv preprint arXiv:2505.14910 (2025).

 [2]  Dai, Shuqi, et al. "Expressivesinger: Multilingual and multi-style score-based singing voice synthesis with expressive performance control." Proceedings of the 32nd ACM International Conference on Multimedia. 2024.

 [3]  Dai, Shuqi, et al. "Everyone-Can-Sing: Zero-Shot Singing Voice Synthesis and Conversion with Speech Reference." ICASSP 2025-2025 IEEE International Conference on Acoustics, Speech and Signal Processing (ICASSP). IEEE, 2025.

**Questions:**

See Weakness section above.

---

### Official Review · Reviewer_dRYX · 2025-11-01

**Soundness:** 2
**Presentation:** 2
**Contribution:** 2
**Rating:** 2
**Confidence:** 3

**Summary:**

This paper formalizes Non-Human Singing Generation (NHSG) by introducing two tasks: Non-Human Singing Voice Synthesis (NHSVS) and Conversion (NHSVC). The authors propose CartoonSing, a unified two-stage framework bridging human and non-human timbres. Stage 1 trains a score representation encoder on human singing to predict frame-level content tokens (multi-layer K-means quantized SSL features) and F0 trajectories; Stage 2 uses a timbre-aware vocoder conditioned on tokens, pitch, and timbre embeddings to synthesize waveforms, trained jointly on human and non-human audio with an unpaired timbre-conditioning strategy and auxiliary predictor losses. Experiments across Chinese/Japanese songs and instrumental/bird/general timbres show CartoonSing improves non-human timbre similarity over VISinger2 and Samoye while retaining pitch and timing, with domain-specific finetuning further stabilizing outputs. Ablations highlight the importance of disentangled content tokens and timbre encoders. The work provides datasets, metrics, and reproducibility details, positioning NHSG as a practical, scalable path to creative singing beyond human timbres.

**Strengths:**

1. This paper formally defines NHSVS/NHSVC and unifies SVS/SVC for zero-shot generation beyond the human timbre manifold.
2. Comprehensive training and evaluation. broad datasets (22 singing, 10 non-human), mixed objective/subjective metrics, competitive baselines, detailed ablations on content/timbre representations, and reproducibility commitments with public audio demos.

**Weaknesses:**

1. This is a very new task, so baseline comparison is a bit meaningless. The baselines (VISinger 2 and SaMoye-SVC) are extremely weak under the experimental setup of the authors, so I cannot know how "good" the model is. The only way to perceive the model's performance is the demo page. However, even though the SIM metrics of the proposed model has reached the best, the generated voices still sounds very unpleasant. The authors may need to conceive a better metric to measure the quality of the generated audio.
2. Continuation of Weakness 1: the proposed model has higher SIM scores, but, subjectively speaking, the voice generated is less pleasant than that of the baselines. This makes me wonder: do we really need a violin to sing poorly, just to be more like a human? This also leads to a new problem: the authors cannot control the extent to which the voice generated resemble the target musical instruments.
3. The authors used Japanese datasets for training and evaluation, but there is no Japanese songs on the demo page. Also, MOS-T is not conducted on Japanese generations. I wonder why.
4. According to the metric LF0 RMSE, the proposed model has worse pitch accuracy. There is no discussion about that.

**Questions:**

1. The authors mentioned that the content tokens are timbre-disentangled (line 258). How to prove that? Training with two separate inputs doesn't necessarily make them disentangled.
2. The authors only use Hubert as an alternative. Which version of Hubert? Also, why? There are many more semantic encoders such as WavLM [1], GLM4-Voice [2], CosyVoice 2 [3], etc. Many of them claim to be timbre-disentangled. Why not try those?

References
[1] WavLM: Large-Scale Self-Supervised Pre-Training for Full Stack Speech Processing, https://arxiv.org/abs/2110.13900.
[2] GLM-4-Voice: Towards Intelligent and Human-Like End-to-End Spoken Chatbot, https://arxiv.org/abs/2412.02612.
[3] CosyVoice 2: Scalable Streaming Speech Synthesis with Large Language Models, https://arxiv.org/abs/2412.10117.

**Details Of Ethics Concerns:**

No concerns.

---

### Note · Authors · 2025-12-04

I have read and agree with the venue's withdrawal policy on behalf of myself and my co-authors.